# Evaluation of a Method for Calculating the Height of the Stable Boundary Layer Based on Wind Profile Lidar and Turbulent Fluxes

**Haijiong Sun** [1,2], **Hongrong Shi** [3,*], **Hongyan Chen** [1], **Guiqian Tang** [1], **Chen Sheng** [2,4], **Ke Che** [2,3] and **Hongbin Chen** [3]

1   State Key Laboratory of Atmospheric Boundary Layer Physics and Atmospheric Chemistry, Institute of Atmospheric Physics, Chinese Academy of Sciences, Beijing 100029, China; sunhaijiong@mail.iap.ac.cn (H.S.); chenhy@mail.iap.ac.cn (H.C.); tgq@mail.iap.ac.cn (G.T.)
2   The State Key Laboratory of Numerical Modeling for Atmospheric Sciences and Geophysical Fluid Dynamics, University of Chinese Academy of Sciences, Beijing 100049, China; shengchen@lasg.iap.ac.cn (C.S.); cheke@mail.iap.ac.cn (K.C.)
3   Key Laboratory of Middle Atmosphere and Global Environment Observation, Institute of Atmospheric Physics, Chinese Academy of Sciences, Beijing 100029, China; chb@mail.iap.ac.cn
4   State Key Laboratory of Numerical Modeling for Atmospheric Sciences and Geophysical Fluid Dynamics (LASG), Institute of Atmospheric Physics, Chinese Academy of Sciences, Beijing 100029, China
*   Correspondence: shihrong@mail.iap.ac.cn

**Abstract:** The height of the stable boundary layer (SBL), known as the nocturnal boundary layer height, is controlled by numerous factors of different natures. The SBL height defines the state of atmospheric turbulence and describes the diffusion capacity of the atmosphere. Therefore, it is unsurprising that many alternative (sometimes contradictory) formulations for the SBL height have been proposed to date, and no consensus has been achieved. In our study, we propose an iterative algorithm to determine the SBL height $h$ based on the flux–profile relationship using wind profiles and turbulent fluxes. This iterative algorithm can obtain temporally continuous, accurate estimates of $h$ and is widely applicable. The predicted $h$ presents relatively good agreement with four observation-derived SBL heights, $h_J$, $h_1$, $h_i$, and $h_\theta$ ($h_J$: maximum wind speed height, $h_1$: zero wind shear height, $h_i$: temperature inversion height, and $h_\theta$: height at which 0.8 times the inversion strength appears for the first time), especially with $h_\theta$, which shows the best fit. In addition, $h$ exhibits a low absolute difference and relative difference with $h_J$, which presents the second-best result. The agreement with $h_i$ and $h_1$ may be satisfactory, but small differences are observed, and the one standard deviation of the mean relative difference is large. In addition, the predicted $h$ is compared with other SBL height estimation methods, including diagnostic, $\lambda_1$, $\lambda_2$ and $\lambda_3$ (three typical dimensional scale height parameters) and prognostic equation-based methods, $\lambda(h)$ (an equation for the growth of $h$ developed). The diagnostic formulas are found to be appropriate, especially under extremely stable conditions. Additionally, the equation of $\lambda_3$ presents the best result of all the dimensional scale height parameters. However, the prognostic equation $\lambda(h)$ in our study is very unsatisfactory.

**Keywords:** stable boundary layer; flux–profile relationship; wind profiles; turbulent fluxes

## 1. Introduction

The atmospheric boundary layer (ABL) plays an important role in the whole atmospheric system, as it regulates the exchanges of heat, moisture, and momentum between the Earth's surface and the free atmosphere [1–3]. The ABL height must be known for a number of practical applications, for example, when modeling the dispersion of pollutants, where the upper boundary of the turbulent layer plays a role as an impenetrable barrier to pollutants released at the surface [4–7].

The daytime convective boundary layer involves far more complex physical processes with intense turbulence and, thus, is more difficulty to measure or compute than its nighttime counterpart [8]. In contrast, the turbulence in the nocturnal boundary layer is suppressed due to stable stratification. The stable boundary layer (SBL) is easier to explore the physical structure; nevertheless, many uncertainties remain. Over the past few decades, numerous measurement techniques have been proposed to estimate the SBL height ($h$) based on turbulent parameters (fluxes, turbulent kinetic energy, Richardson number) or vertical profiles of atmospheric parameters (temperature, humidity, wind, aerosol concentration and optical (thermal) turbulence estimated from recorded phase fluctuations) [7,9,10]. For example, Lenschow et al. [11] and Caughey et al. [12] defined $h$ as corresponding to the height at which the turbulent kinetic energy (TKE) drops to 5% of its surface value. Unfortunately, vertical profiles of TKE are difficult to measure due to the lack of appropriate instruments, many of which are not able to determine $h$. Banta et al. [13,14] used profiles calculated from high-resolution Doppler lidar (HRDL) scanning data to obtain the streamwise mean wind $U(z)$ and variance $\sigma_u^2(z)$, the latter of which has been shown to be approximately equal to TKE under stable conditions [15,16] insomuch that profiles of $\sigma_u^2$ are essentially equivalent to TKE profiles in the SBL. Alternatively, the SBL has been defined as the height where the magnitude of the momentum flux reduces to 1% of its surface value [17]. However, few people can obtain accurate measurements of the vertical velocity at $h$ in practical applications.

Some researchers determined $h$ based on vertical profiles of temperature, humidity, and wind, e.g., in [18–26]. However, these approaches suffered from many shortcomings, such as crossing the SBL along a slanted path within a few minutes, providing a "snapshot"-like profile, having a limited height resolution of routine ascents, the impossibility of obtaining measurements under high wind speeds, noncontinuous observations, and a lack of turbulent fluxes [27]. In addition, the height of the low-level jet (LLJ) maximum calculated by Doppler lidar can sometimes serve as a good estimate of $h$ during the nighttime [18,27–29]. However, the LLJ does not typically occur during the entire observation period, and continuous observations cannot be made. In addition to various measurement methods, many computational parameterizations of $h$ have been proposed in the literature [22,30–38]. However, there are controversial debates on which expression is the most suitable for determining $h$ [31,39–41].

The different methods of previous studies have provided a variety of ways to estimate $h$. However, all of these approaches have shortcomings. As an alternative, Zilitinkevich et al. [42] proposed a theoretical algorithm to determine $h$ based on the surface-layer flux–profile relationship. The advantage of this algorithm is its combination of wind profiles and surface fluxes. Zilitinkevich et al. [42] verified this height-determination algorithm by using a database computed with a large-eddy simulation, but the algorithm was not tested under the conditions of an actual situation. Further work using measured data is carried out in our study. By using observational data detected by Doppler wind lidar and an ultrasonic anemometer, we have improved the method of Zilitinkevich et al. [42] to obtain continuous and accurate estimates of $h$, and the improved approach is easier to popularize. We also evaluate our improved method with diagnostic equations and observation-derived data. The observation sites, measurements, and data processing are described in Section 2. The calculation method and comparison results are presented in Section 3. Section 4 gives the conclusions and discussion.

## 2. Sites, Synoptic Condition, Instruments, and Data Processing

### 2.1. Sites, Synoptic Condition

An intensive field campaign was conducted during the summertime (4–6 August 2020) in Xilin Gol League, Inner Mongolia, China, to measure the aerosol-cloud-boundary layer interaction; the vegetation in the study region is dominated by grasses, and there are no residential or industrial areas near this field (Figure 1, 42°11′ N, 114°56′ E). Synchronous

measurements were performed at Site A (1280 m ASL) and Site B (1274 m ASL). The straight-line distance between the two sites is approximately 500 m.

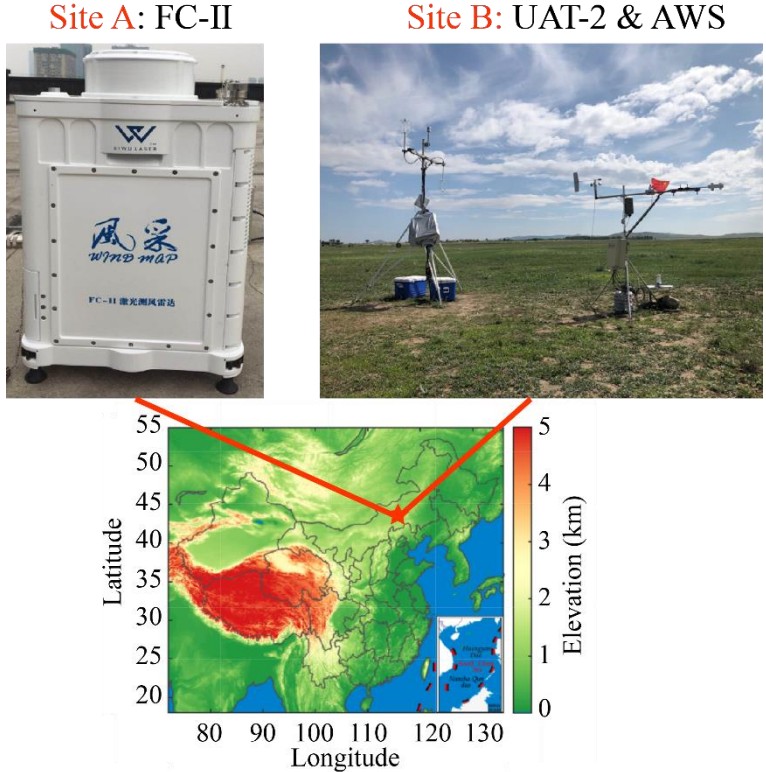

**Figure 1.** Topographic map of the experimental site. The straight-line distance between the two sites is approximately 500 m. The portable Doppler wind lidar (FC-II) was deployed at Site A. The ultrasonic anemometer thermometer (UAT-2) was deployed at Site B. The location of the field experiment was at the southern edge of Xilin Gol League (pentagram in the button map).

*2.2. Instruments*

Synoptic-scale weather patterns during our observation period reveal that the experiment site was mainly controlled by high atmospheric from Mongolia. Figure S1a–d show that the north-west Mongolia high-pressure system was over our experiment site at 02:00 BT and 05:00 BT (Beijing time) 5–6 August, indicating that downward air motion suppressed air mass from the ground to the troposphere (Figure S1).

2.2.1. Portable Doppler Wind Lidar

The FC-II portable Doppler wind lidar (Norinco Group, Beijing, China) deployed at Site A uses a narrow-linewidth pulsed laser as the emission source with the Doppler coherence detection principle (Figure 1, top left corner). The FC-II provides horizontal and vertical winds $U(z)$ with a vertical resolution of 50 m and a temporal resolution of 3 s. The maximum measured height and uncertainty depend on the environmental and weather conditions, such as aerosol backscattering, turbulence, humidity, and precipitation [43]. Details of the operational parameters are provided in Supplementary Information Table S1. More information about the FC-II and the data quality control procedure can be found in the literature [44].

2.2.2. Ultrasonic Anemometer Thermometer

An ultrasonic anemometer thermometer-II (UAT-2, Chinese Academy of Sciences, in Beijing, China) was deployed at Site B (Figure 1, top right corner). The UAT-2 uses an array of transducers arranged on nonorthogonal axes. Three transducer pairs compose three sonic paths oriented at an elevation angle of 45° to the horizontal plane; there is an azimuth

angle of 120° between each path, and the path length between transducers is 15 cm. The sampling frequency can reach 100 Hz. The additional operating parameters of the UAT-2 can be found in Table S1b. More information regarding the UAT-2 can be found in [45]. The methods used to calculate the velocity ($U$) and virtual potential temperature ($\theta_v$) can be found in S1. The data obtained by the UAT-2 are characterized by a high frequency and large variation. To ensure the reliability of the data, the outliers and missing records are replaced by the quintuple variance ($5\sigma^2$). Based on Grubbs criterion, if the residual corresponding to a measured value exceed $5\sigma^2$, the data should be omitted. A second iteration is proceeded by the same $5\sigma^2$ method until all the outliers are eliminated [46]. Figure S2 shows the three calculated wind components ($u$, $v$, $w$) after quality controlling the data, and the velocity shows a good fit with the data measured by an automatic weather station (AWS) (Figure S2d, red solid line). The methods used to calculate the turbulent fluctuations and fluxes can be found in S2.

### 2.2.3. Other Observational Data

Meteorological variables (temperature $T$, pressure $P$, relative humidity RH, wind speed WS, and wind direction WD) were synchronously measured by an AWS (Figure 1, Site B, bottom right corner) with four-cup anemometers (Mode: 034, Met One Instruments, USA), a standard meteorological probe (Mode: HC2A-S3, Rotronic, NE, USA), and a barometric pressure sensor (Mode: CS106, Vaisala, Finland) with a temporal resolution of 1 s [47]. In addition, sounding instruments (Mode: RS92, Vaisala, Finland) were mounted on an unmanned aerial vehicle (UAV) with multiple sensors for air pressure, temperature, and relative humidity measurements (Site B). The detailed sampling setup and calibration of the sounding instruments are described in [48]. An additional comparison of the potential temperature ($\theta$) and water vapor mixing ratio ($r$) between the sounding data from the balloon and UAV is provided in Figure S3. The two instruments show similar patterns, with mean biases for $\theta$ and $r$ of 0.28 K and 1.1 g kg$^{-1}$, respectively. Both sounding sensors capture the thermodynamic structures of the SBL reasonably well.

### 2.3. Determination of Stationary and Nonstationary Conditions

As reported in previous studies, the SBL is defined as the stability parameter $\xi$, which is positive ($\xi = \frac{z}{L_{MO}}$, where $L_{MO}$ is the Monin–Obukhov length). To ensure the reliability of the data in the study period, we applied a stricter method from Mahrt et al. [49] to discriminate between stationary and nonstationary periods. Mahrt et al. [49] defined the following ratio:

$$\beta \equiv \frac{\left(\sigma_u^2 + \sigma_v^2\right)^{1/2}}{\overline{U}} \tag{1}$$

where the standard deviations are computed from the six 10 min averages of the wind components measured by an ultrasonic anemometer for a 1 h period and $\overline{U}$ is the 1 h averaged wind speed. Then, the hourly values of $\beta$ are averaged over the records. The record is classified as stationary if $\beta$ is less than 0.1. The local averaging length $L$ must be chosen to be sufficiently large ($L$ = 1 h is appropriate) so that the perturbation flow includes most of the turbulence. Figure 2 shows the time series of $\beta$ during the observed period. We focused mainly on stationary conditions, which all appeared at nighttime, and four episodes (Ep. 1: 00:00–07:00 LST on 4 August; Ep. 2: 16:00 LST on 4 August to 08:00 LST on 5 August; Ep. 3: 00:00–08:00 LST on 6 August; Ep. 4: 18:00 LST on 6 August to 00:00 LST on 7 August) were determined.

Figure 3 shows the momentum flux variations as a function of the Reynolds averaging scale ($L$) for both stationary and nonstationary episodes. For the stationary episodes, when $L \geq 60$ min, most of the turbulence flux appears to be captured (Figure 3a). Conversely, the momentum flux tends to increase as $L$ increases for the nonstationary episodes (Figure 3b) and does not reach a constant value with increasing $L$. This period appears nonstationary and is thus excluded from our study. We chose $L$ = 60 min in our following analyses.

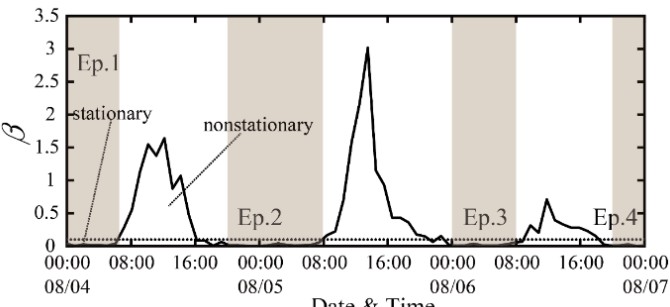

**Figure 2.** Hourly averaged values of $\beta$ from 4–6 August 2020. The dotted line represents $\beta = 0.1$. If $\beta$ exceeds 0.1, this episode is classified as nonstationary. Figure 1. Ep. 2, Ep. 3, and Ep. 4 were determined by $\beta < 0.1$; these episodes are marked as having a brown shaded background. A distinct stratification emerges with a dividing line of $\beta = 0.1$.

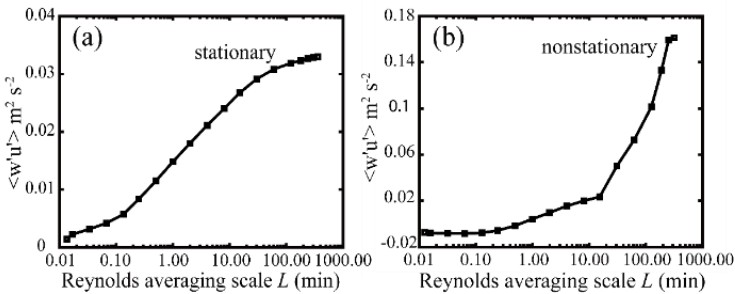

**Figure 3.** Dependence of the momentum flux on the Reynolds averaging scale $L$ used to compute the perturbation quantities for (**a**) stationary and (**b**) nonstationary episodes.

## 3. Calculation Method and Comparison Results

### 3.1. Using fluxes and Wind Profiles to Calculate h

Zilitinkevich et al. [42,50] proposed a diagnostic equation in which the squared reciprocals of $h$ were satisfied with some second-order linear interpolation terms using a large-eddy simulation (LES) database on a stable and neutral atmospheric boundary layer:

$$\frac{1}{h^2} = \frac{f^2}{C_R^2 \tau_*} + \frac{N|f|}{C_{CN}^2 \tau_*} + \frac{|f\beta_b F_*|}{C_{NS}^2 \tau_*^2}$$

(2)

where $C_R = 0.6$, $C_{CN} = 1.36$, and $C_{NS} = 0.51$ are empirical dimensionless constants, $f$ is the Coriolis parameter, and $N$ is the Brunt–Vaisala frequency (typically $N \sim 10^{-2}$ s$^{-1}$). The right-hand terms represent (from left to right) 'true neutral' (TN), 'conventionally neutral' (CN) and 'nocturnal stable' (NS). These three types of neutral and stable ABLs are distinguished when $B_s = 0$, $N = 0$; $B_s = 0$, $N > 0$ and $B_s < 0$, $N = 0$, where $B_s$ is the buoyancy flux at the surface.

From Equation (2), given $\tau_*$ and $F_*$ calculated by the UAT-2 data, the calculated value of $\tau_*$ and $F_*$, and $N$ achieved by some other means, $h$ can be determined. As mentioned above, Equation (2) is a linear interpolation of terms. At the very beginning, assuming the CN term was not taken into account ($N = 0$), $h = h_a$ can be estimated by Equation (2) (Figure 4, Step 1). $h_a$ was the height containing TN and NS. In order to obtain integrated SBL height that contains the CN term, further calculations were made.

$\tau$ and $F_\theta$ can be calculated at multiple heights by using quasi-universal dependencies [50,51]:

$$\frac{\tau}{\tau_*} = \exp\left[-\frac{8}{3}\left(\frac{z}{h}\right)^2\right]$$

(3)

$$\frac{F_\theta}{F_*} = \exp\left[-2\left(\frac{z}{h}\right)^2\right]$$

(4)

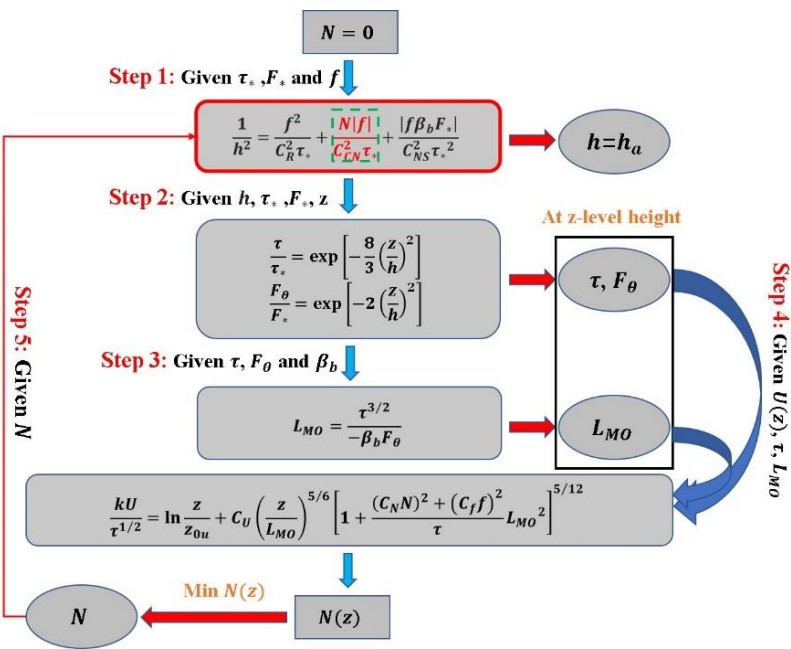

**Figure 4.** Flow chart for calculating $h$ including 5 steps. Step 1: assuming $N = 0$, $h_a$ is calculated with given $\tau_*$, $F_*$ and $f$; step 2: $\tau$ and $F_\theta$ at each z-level height (z = 50, 100, . . . , 400 m) are estimated with $h_a$, $\tau_*$, $F_*$ and $z$; step 3: calculating $L_{MO}$ at z- level heights using $\tau$, $F_\theta$ and $\beta_b$; step 4: based on $U(z)$, $\tau$ and $L_{MO}$, calculating $N$ at z-level heights and taking the minimum value of $N(z)$; step 5: substitute $N$ into Equation (2) in the first step, and the $h$ can be determined ultimately.

$z$ is equal to the heights of the wind speed profile $U(z)$. Given the surface fluxes $\tau_*$ and $F_*$ and the estimated $h_a$, $\tau$ and $F_\theta$ can be calculated at z-level heights by using Equations (3) and (4), respectively (Figure 4, Step 2).

Moreover, M-O length scale $L_{MO}$ can be determined based on Monin–Obukhov (M-O) theory (Figure 4, Step 3). Monin–Obukhov (M-O) theory states that the turbulent regime in the stratified surface layer is fully characterized by the turbulent fluxes, $\tau \approx \tau_*$ and $F_\theta \approx F_*$ ($\tau \approx u_*^2$ and $F_\theta = \frac{g}{\theta}\overline{w'\theta'}$, details seen in S2), and the buoyancy parameter, $\beta_b = g/T_0$ (where $g$ is the acceleration due to gravity and $T_0$ is a reference value of the absolute temperature), which determine the M-O length scale when given $\tau$ and $F_\theta$ at z-level heights:

$$L_{MO} = \frac{\tau^{3/2}}{-\beta_b F_\theta} \tag{5}$$

Zilitinkevich and Esau [50,52] received analytical approximations of the mean wind based on the flux–profile relationship:

$$\frac{kU}{\tau^{1/2}} = \ln\frac{z}{z_{ou}} + C_U\left(\frac{z}{L_{MO}}\right)^{5/6}\left[1 + \frac{(C_N N)^2 + \left(C_f f\right)^2}{\tau}L_{MO}^2\right]^{5/12} \tag{6}$$

where $C_N b$ = 0.1, $C_f$=1 and $C_U$ = 3 are empirical dimensionless coefficients, $z_{ou}$ is the aerodynamic roughness length for momentum, and the von Karman constant is $k$ = 0.4. Given the wind profile $U(z)$ measured by the FC-II, the turbulent fluxes $\tau$, and the M-O length $L_{MO}$, at the computational level z ($h_a < Z < 2h_a$), the free-flow Brunt–Vaisala frequency $N(z)$ can be determined (Figure 4, Step 4). Then, a unique $N$ with a minimum value of $N(z)$ can be identified. Finally, $N$ is substituted into Equation (2), and the SBL height $h$ is determined (Figure 4, Step 5).

In summary, we propose an improved iteration method to calculate $h$ based on the work of [6,42,50]; this method requires both turbulent fluxes $\tau$ and $F_\theta$ at different

heights and the wind speed profile $U(z)$. By calculating the surface-layer turbulent fluxes $\tau_* = \tau|_{z=0}$ and $F_* = F_\theta|_{z=0}$, the profile of the average wind speed $U(z)$ can be substituted into an iterative algorithm to determine $h$. Figure 5 shows the hourly averaged values of $h$ calculated by above method.

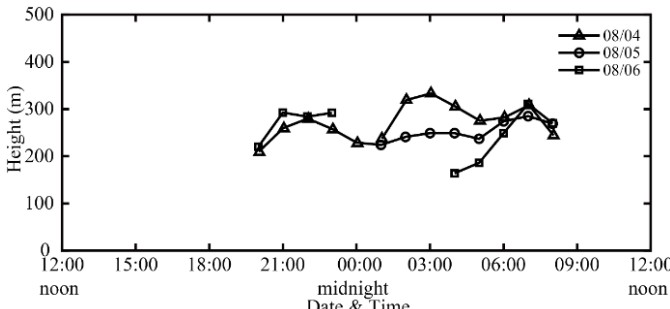

**Figure 5.** Hourly averaged values of $h$ from 00:00 on 3 August 2020 to 00:00 on 7 August 2020.

Many physical factors restrict the value of $h$. We considered three physically essential ABL height formulations by linear interpolation: TN, CN, and NS (see Equation (2)). The contributions of these three basic regimes to $h$ were interpolated by using the data of our study. The TN term accounts for 66%~97%, indicating that the 'true neutral' boundary layer is the main factor. In addition, the CN term accounts for 0% to 16%, the lowest contribution term of the three regimes, whereas the values of $N^2$ are $1 \times 10^{-4}$~$5 \times 10^{-4}$ s$^{-1}$, indicate a stable atmospheric state. The contribution of the NS term ranges from 3% to 28%, indicating a relatively important effect of $h$ that cannot be ignored. Equation (2) is a synthesis equation that integrates the TN-CN factors (where $B_s = 0$) and TN-NS factors (where $N = 0$). Other studies, for example, [53], did not distinguish between the TN and CN boundary layers, and neither did [54], who derived the expression $u_*|fN|^{-1/2}$ for the maximal depth of the oceanic upper mixed layer. Further comparison with other formulations is presented in Section 3.2 to discuss their differences. Nevertheless, Equation (6) gives the flux–profile relationship that is suitable for practical applications, as this equation clarifies the characteristic function $\Psi_U = \frac{kU}{\tau^{1/2}} - \ln\frac{z}{z_{0u}}$ based on similarity theory and a dependence on $\zeta$ that can be accurately approximated by the power law $\Psi_U = C_U(\xi)^{5/6}$. Moreover, the second term in the square bracket of Equation (6) is usually small compared to the first term. The numerical solution of $U(z)$ can be simplified by the fact that the major terms on the right side are a logarithmic term and an exponential term. Hence, a comparison with the calculated $U\_cal$ and the values of the wind profile $U\_real$ measured by the FC-II at computational level $z$ is presented in Figure S4. The data were chosen from 50 to 400 m, for which the vertical resolution is 50 m, and the strong correlation ($R^2 = 0.91$) indicates a good fit with the calculated values below 400 m. In particular, below 300 m, the wind speeds are generally less than 9 m s$^{-1}$. This result further confirms the feasibility of the proposed flux–profile relationship of Equation (6). Above 300 m (>9 m s$^{-1}$, Figure S4), however, the wind speeds do not seem to fit well, although a relatively uncertain wind speed has little effect on obtaining $N$ because the second term in the square bracket of Equation (6) is small, as mentioned above.

### 3.2. Compared with Other Predicted SBL Heights

We compared the $h$ values determined in Section 3.1 with the SBL heights predicted from three different dimensional scale height parameters (Equations (7)–(9)). Clarke [55] speculate that based on M-O similarity theory, the SBL height $h$ may be determined by a dimensional scale height parameter:

$$\lambda_1 \equiv h \propto k u_* / f \tag{7}$$

Alternatively, Deardorff [56] suggested an empirical formula for determining $h$:

$$\lambda_2 \equiv h \propto \left( \frac{1}{30L_{MO}} + \frac{f}{0.35u_*} \right)^{-1} \tag{8}$$

Furthermore, Businger and Arya [17] deduced a formula through a theoretical steady-state model:

$$\lambda_3 \equiv h \propto (ku_* L_{MO}/f)^{\frac{1}{2}} \tag{9}$$

The relationships given in Equations (7) and (9) are diagnostic in nature. However, $\lambda_1$ is actually the first term on the right side of Equation (3). In addition, $\lambda_2$ reflects the interpolation between the reciprocals of the two scales and characterizes the stabilizing effect of local buoyancy forces on turbulence and the effect of the Earth's rotation; different from Equation (3), Equation (8) depends on the chosen order of each term in the linear interpolation. Finally, Equation (9) is deduced based on similarity theory, similar to Equation (7).

The SBL height data are classified into four categories, namely, extremely, very, moderately, and slightly stable (see Table S3), by computing the dimensionless stability parameter (see Table S3). The calculated values of $h$ and those of the SBL heights predicted by the dimensional parameters $\lambda_1$, $\lambda_2$ and $\lambda_3$ under these four stability classifications are plotted in scatter diagrams in Figure 6. In general, $\lambda_1$ is less consistent with $h$ than are $\lambda_2$ and $\lambda_3$. Figure 6b shows that $\lambda_2$ is underated with respect to $h$ in comparison with the observation-derived SBL height under all four stability classifications. As shown in Figure 6c, the $h$ values predicted by the diagnostic formula of $\lambda_3$ present the best fit with the calculated $h$ (further details will be presented in Section 3.2).

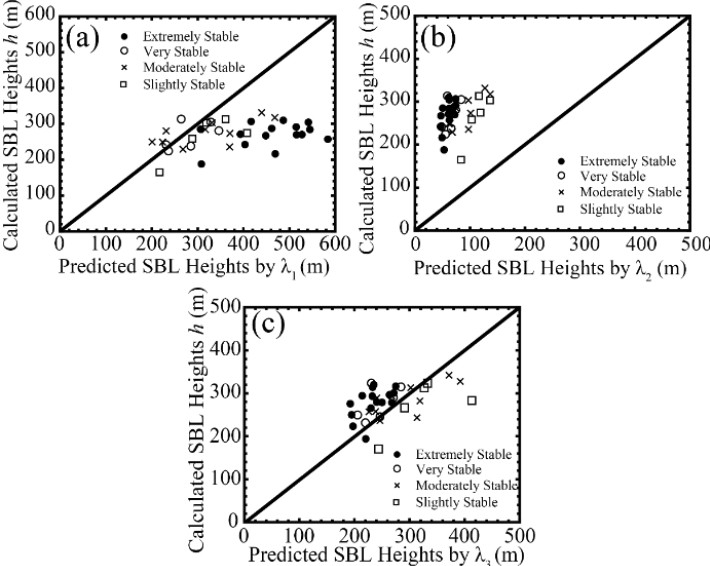

**Figure 6.** Scatter diagrams of the predicted versus the calculated heights of the stable boundary layer: (**a**) $\lambda_1 = ku_*/f$ vs. $h$, (**b**) $\lambda_2 = \left( \frac{1}{30L_{MO}} + \frac{f}{0.35u_*} \right)^{-1}$ vs. $h$, and (**c**) $\lambda_3 = (u_* L_{MO})^{\frac{1}{2}}$ vs. $h$.

The correlation coefficients between $h$ and the diagnostic formulas of the dimensional scale height parameters $\lambda_1$, $\lambda_2$, and $\lambda_3$ under each stability category are shown in Table 1. As expected, the correlation coefficients calculated for the extremely stable case exhibit the largest values among all the stable classifications ($n = 36$), followed by those calculated for the very stable case. This is consistent with the fact that these diagnostic formulas are based on M-O similarity theory, which holds well under extremely stable conditions [56]. Moreover, the $r$ values increase as the stability increases (Table 1), indicating that the diagnostic formulas become increasingly useful with greater stability. This occurs presumably because under extreme conditions, the nocturnal boundary layer heights are very shallow, generally

lower than 200 m, with the least scatter for the cases considered. As a consequence, the correlation coefficients are high under extremely stable conditions.

**Table 1.** Correlation coefficients (*r* values) between *h* and the diagnostic height parameters. The corresponding Z values are in parentheses.

| Correlation Coefficients | $r(h,\lambda_1)$ | $r(h,\lambda_2)$ | $r(h,\lambda_3)$ |
|---|---|---|---|
| (I) Slightly stable $27 > \mu_0 > 5$ n = 5 | 0.36 (1.31) | 0.45 (2.14) | 0.53 (2.04) |
| (II) Moderately stable $45 > \mu_0 > 27$ n = 10 | 0.56 (1.10) | 0.66 (1.10) | 0.54 (1.10) |
| (III) Very stable $78 > \mu_0 > 45$ n = 6 | 0.65 (1.47) | 0.74 (1.97) | 0.61 (2.35) |
| (IV) Extremely stable $\infty > \mu_0 > 78$ n = 15 | 0.78 (2.04) | 0.88 (1.98) | 0.65 (2.12) |
| (V) Total cases $\infty > \mu_0 > 5$ n = 36 | 0.36 (2.17) | 0.41 (2.50) | 0.46 (2.86) |

Table 1 shows the results for all 36 h of data. The *r* values range between 0.36 and 0.88. This indicates that linear relationships between *h* and the dimensional height parameters may be satisfactory but may exhibit considerable scatter. Moreover, these three diagnostic formulas yield values comparable to *h*, although under certain stability conditions, one formulation may perform slightly better than the others. For example, the diagnostic formula of $\lambda_2$ under extremely stable conditions (*r* = 0.88) performs better than those of $\lambda_1$ (*r* = 0.78) and $\lambda_3$ (*r* = 0.65), although both of the latter two formulas yield high correlation coefficients. Hence, a statistical test for the significance of these correlation coefficients may be useful by using the Z values (see S4) bracketed and listed in Table 1. It is obvious that for all the cases considered (n = 36), there may be a relatively high correlation between the calculated *h* and the values predicted by the three diagnostic formulas. However, a careful inspection of the calculated Z values for each stability class indicates that these correlations may not be significant. For example, the value of $r(h, \lambda_1)$ under very stable conditions shows a relatively high *r* value of 0.65. However, the Z value of $\lambda_1$ (Z = 1.47) is less than 1.96, indicating that this correlation coefficient is not reliable. In general, although $\lambda_1$ shows less consistency with *h* (Figure 6a), there is reason to believe that all three diagnostic formulas could provide good fits with *h* under extremely stable conditions, especially those of $\lambda_2$ and $\lambda_3$.

In general, $u_*$ and $L_{MO}$ usually remain nearly constant, and the SBL height thus determined will remain independent of time [57]. However, *h* varied dramatically over time during our experiment. Thus, we use the prognostic equation for the growth of *h* developed by [21]:

$$\lambda(h) = \frac{\partial h}{\partial t} = 250u_*[1 - h/(0.35u_*/f)] \tag{10}$$

If the initial *h* is known, the prognostic equation of Equation (10) will reveal the evolution of the stable boundary layer.

The correlation coefficients (*r*) between the predicted SBL heights $\lambda(h)$ and the changes in the calculated SBL height $\Delta h$ at a 1 h interval are plotted in Figure 7. $\Delta h$ performs very inconsistently with the predicted SBL heights. The correlation coefficients between $\Delta h$ and $\lambda(h)$ are nearly zero (Table S2, $r[\Delta h, \lambda(h)] = 0.02$), indicating that the above prognostic equation cannot accurately determine the changes in the SBL height. However, the dimensional scale height parameters ($\frac{\partial \lambda_2}{\partial t}$ and $\frac{\partial \lambda_3}{\partial t}$) show higher *r* values (0.59 and 0.64, Table S2). Based on tests on the statistics concerning the sample correlation coefficients (see S4), the Z

values of ($\Delta h$ and $\frac{\partial \lambda_2}{\partial t}$) and ($\Delta h$ and $\frac{\partial \lambda_3}{\partial t}$) shown in Table S2 indicate that the change in the SBL height is significantly correlated with both $\frac{\partial \lambda_2}{\partial t}$ and $\frac{\partial \lambda_3}{\partial t}$. In summary, the prognostic formulas are more suitable for $\lambda_2$ and $\lambda_3$, but their height changes seem nonsignificant.

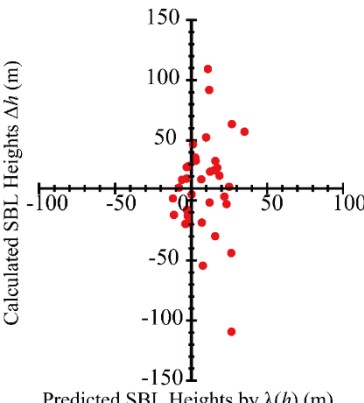

**Figure 7.** Scatter diagram of the predicted $\lambda(h)$ versus the calculated height changes $\Delta h$ in the stable boundary layer.

### 3.3. Comparison with the Observation-Derived SBL Heights

### 3.3.1. SBL Heights Derived from Wind Profiles

In previous studies, wind profiles were used to determine the SBL heights [58], which are usually defined as the height of the maximum wind speed $h_J$ or the height of the zero value in the wind shear profile $h_1$. Here, the wind profiles obtained from the FC-II available at 30 s intervals are averaged over 10 min. The profile shapes can be grouped into three distinct types: Type I (Figure S5a), the classic LLJ shape with a distinct maximum or "nose"; Type II (Figure S5b), a uniform or "flat" profile; and Type III (Figure S5c), a profile with a layered structure. In most cases, the height of the top layer with a layered structure (Figure S5c) was chosen as $h_J$. A few profiles with shapes that do not fit among these three types were not taken into account in our study. In addition, for the shape of Type II that the wind speed was relatively uniform or "flat" with a deep layer overlying the layer of strong surface-based shear (Figure S5b). Although this type of profile may not be defined as an LLJ (Type I) because of the lack of a distinct nose, for Type I and Type II, the $h_J$ values could be determined by the same way (e.g., the $h_J$ values are consistent with the $h_1$ values shown in Figure S5a,b). These type profiles are the most common type of profile overall, occurring in nearly 96% of the total (a total of 180 runs). The wind shear magnitude below the height of $h_1$ is relatively invariant with a value of ~0.04 s$^{-1}$ during the nighttime (Figure S5). These near-neutral lapse rates indicate steady atmospheric stratification. Thus, we believe that these modest wind shear values could be related to $h$.

The time series of the $h$ calculated in Section 3.1 are compared with the maximum wind speed height $h_J$ and the zero-wind shear height $h_1$. Figure 8 shows a time–height cross section of the 10 min wind profile maximum height $h_J$, wind shear height $h_1$, and calculated SBL height $h$ from 4–6 August 2020. Four episodes (Ep. 1, Ep. 2, Ep. 3, and Ep. 4) were chosen (shaded with a brown background, as in Figure 2) and determined by $\beta < 0.1$. Similar variations in $h_J$, $h_1$, and $h$ are presented in Figure 8, although there are a few discrepancies. In particular, Ep. 2 and Ep. 3 show almost the same trend. The correlation coefficient $r(h_J, h)$ is 0.68, which is better than $r(h_1, h)$ with a value of 0.43, indicating a better fit between $h$ and $h_J$. Table 2 shows the first-order linear regression results for all types of $h_J$ for each episode. All four episodes pass the Z-value test except for Type III, which does not fit $h$ well. The regression parameters of Ep. 3, including the correlation coefficient and slope of the regression lines, show the highest values in all four episodes, indicating that the $h_J$ of Ep. 3 fits the $h$ calculated in this study the best, mainly because $h$ is often associated with high wind shear $U(z)$, which is more likely to attenuate the TKE at the height of the wind profile maximum. In contrast, Ep. 2 exhibits relatively

low correlation coefficients for all types. In addition, the mean bias in Ep. 3 exhibits the lowest value, indicating the best fit among the four episodes. In summary, the maximum height of the 10 min profile $h_J$ is well correlated with the calculated value of $h$, and thus, $h_J$ can be considered equivalent to the depth of the SBL in all four episodes, especially Ep. 3.

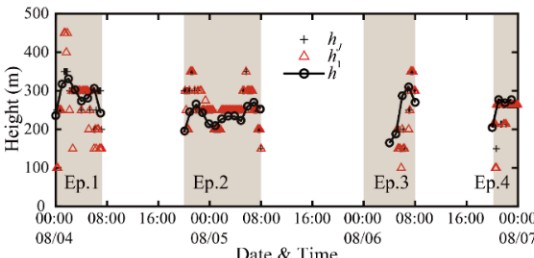

**Figure 8.** Time–height cross section of the 10 min wind profile maximum height $h_J$, wind shear height $h_1$, and calculated SBL height h from 4–7 August 2020. The heights of the maximum wind profile $h_J$ are denoted by black plus signs, the wind shear heights $h_1$ are shown as red triangles, and the black circles with lines indicate the calculated SBL heights $h$. The four episodes (Ep. 1, Ep. 2, Ep. 3, and Ep. 4) are delineated by the brown background (same as Figure 2) and determined by $\beta < 0.1$.

**Table 2.** Correlation coefficients ($R_J$), slopes ($B_J$), and biases ($A_J$) for the linear regressions between $h$ and $h_J$. The regression data are separated for the four episodes (Ep. 1, Ep. 2, Ep. 3, and Ep. 4). The Z-test values are in parentheses.

| Episode | Mean $U(z)$ (m s$^{-1}$) | Type I, II (% & n) | $h = A_J + B_J h_J$ | | |
|---|---|---|---|---|---|
| | | | $A_J$ | $B_J$ | $R_J$ |
| Ep. 1 n = 44 | 3.49 | 95 42 | 154.73 | 0.47 | 0.66 (5.08) |
| Ep. 2 n = 92 | 3.29 | 89 82 | 160.52 | 0.39 | 0.64 (7.15) |
| Ep. 3 n = 16 | 7.78 | 94 15 | 142.09 | 0.51 | 0.76 (3.59) |
| Ep. 4 n = 28 | 4.40 | 96 27 | 373.55 | −0.27 | 0.72 (4.54) |
| Total n = 180 | 4.74 | 92 166 | 140.08 | 0.50 | 0.68 (11.03) |

The differences between $h$ and $h_J$ (e.g., the relative error $|h - h_J|/h$) are given in Figure 9 for all four episodes. The mean relative errors of the 10 min $h_J$ for Ep. 1, Ep. 2, and Ep. 4 are all less than 10%, especially that for Ep. 4 (7.2%). However, the mean relative error (15.0%) for Ep. 3 (Figure 9c) is twice as large as that for Ep. 4. The discrete vertical sampling interval of the FC-II in this study ($\Delta z = 50$ m) could account for this increased mean relative error. Assuming that the vertical sampling interval was small enough, the mean relative error between $h$ and $h_J$ could be improved. The median relative error for Ep. 2 (Figure 9b) is the minimum value of 0.054 among the four episodes. In general, the mean relative errors for all four episodes are relatively small, and the $h_J$ for Ep. 2 is the most consistent with $h$.

### 3.3.2. SBL Heights Derived from Radiosonde Data

Another observation-based method for estimating the SBL height is based on radiosonde profiles of the virtual potential temperature profile ($\theta_v$). The virtual potential temperature at the SBL top height is defined by the following two discriminants:

$$\theta_h = \theta_0 + A\delta \tag{11}$$

$$\delta = \left|\overline{\theta_r} - \theta_0\right| \tag{12}$$

where $h$ is the SBL height and $A$ is an empirical constant (0.8). $\delta$ is the inversion strength defined as the difference between the potential air temperature at the surface ($\theta_0$) and the mean potential temperature in the residual layer ($\overline{\theta_r}$). The value of $\overline{\theta_r}$ is the potential temperature averaged from 400 to 500 m; in our study, the potential temperature in this layer is almost uniform, and the vertical decline rate is taken as <1 K/100 m (for further details, see [59]). In our study, $h_\theta$ is defined as the height where $\theta_h$ appears for the first time. In addition, $h_i$ is defined as the radiosonde-derived height of the temperature inversion. We chose seven vertical profiles from the radiosonde data during three episodes, Ep. 2, Ep. 3, and Ep. 4 (as shown in Figure 10 and Figure S6). All seven profiles were measured during the transition phase (05:00–08:00 LST and 19:00–22:00 LST) of each episode.

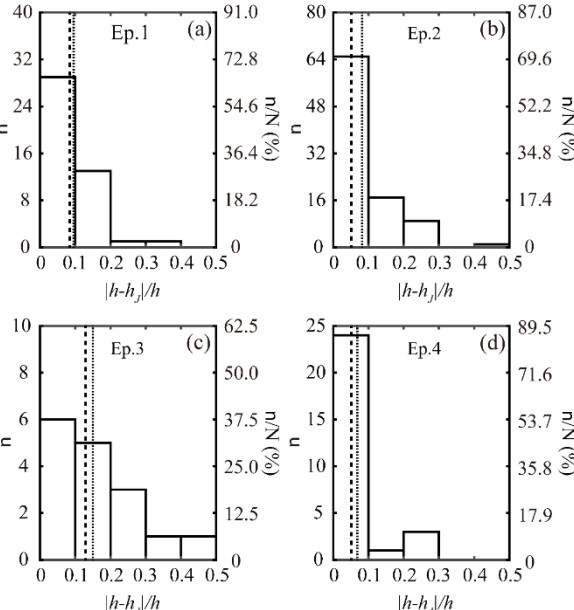

**Figure 9.** Histograms of the normalized absolute errors of the difference between $h$ and $h_J$. All data for the four episodes, (**a**) Ep. 1, (**b**) Ep. 2, (**c**) Ep. 3, and (**d**) Ep. 4, are shown Table S3.

An obvious change in the nocturnal boundary layer height is shown in Figure 10a. The $T$ inversion layer occurred at 20:15 LST, when the observed SBL height was approximately 40 m, and then increased at 05:15 LST the following day. Figure 10a shows that $h_i$ at 05:15 LST on 5 August is characterized by a strong and deep surface inversion (2.4 °C (100 m)$^{-1}$) and then decreased to nearly half (1.3 °C (100 m)$^{-1}$) by 08:00 LST. $h_i$ exhibited a decreasing trend, which accounted for the decreasing intensity of inversion. The changes in $h_i$ were the same as the $h$ variations (see Figure 8) during Ep. 2. Moreover, during the transition phase from approximately 05:00 to 08:00 LST (Figure 10b), the vertical temperature profile decreased from 11 K to 4 K (Figure 10b), which is attributed to surface heating. The variation in $h_\theta$ during the transition phase increased from 175 m to 235 m, which fit well with $h$, and the averaged $h_\theta$ was ~205 m, which was obviously consistent with $h$ (~240 m) (Figure 10b) during Ep. 2. Furthermore, the ground-level water vapor mixing ratio ($r$) increased obviously, indicating an enhancement of surface flux transport caused by the elevation of the SBL. Regarding the performance of $h_\theta$, from all seven runs of data, the relative differences (reDiff) with respect to $h$ (Figure 11b) range between 1% and 22%, and the absolute differences (absDiff) with respect to $h$ (Figure 11a) range between 3 and 50 m. These results indicate that $h_\theta$ is suitable for determining the SBL height in this study. Figure 11b also shows the relative difference with respect to $h_i$, where the mean reDiff value is approximately 36%. This indicates that the correlation between $h_i$ and $h$ may be satisfactory, but there is a considerable difference. In particular, two high values of reDiff, 73% and 53%, appear at approximately 20:15 LST on 4 August and 20:40 LST on 6 August, respectively. The absDiff (<50 m) and reDiff (<22%) of $h_\theta$ are small at these times, as shown

in Figure 11, especially at 08:00 and 20:40 LST on 6 August, when the values are almost the same as *h*.

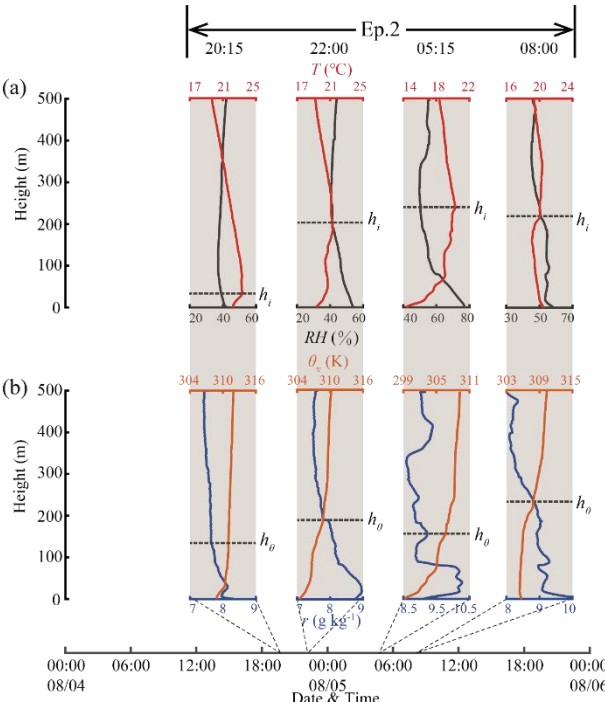

**Figure 10.** Variations in the vertical profiles of the (**a**) temperature (*T*, red solid line) and relative humidity (*RH*, black solid line) and the (**b**) virtual potential temperature ($\theta_v$, orange solid line) and water vapor mixing ratio (*r*, blue solid line) from 00:00 LST on 4 August 2020 to 00:00 LST on 6 August 2020 during Ep. 2. Four times (20:15 LST; 22:00 LST; 05:15 LST; 08:00 LST) were chosen (brown background). The horizontal dotted lines in (**a**) indicate the heights of the inversion layer $h_i$, and those in (**b**) indicate the heights $h_\theta$ where the $\theta_v$ value first exceeds the minimum $\theta_v$ value by 1.5 K.

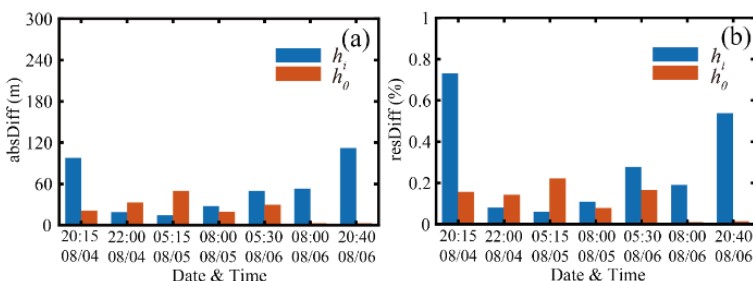

**Figure 11.** (**a**) Absolute differences (absDiff) and (**b**) relative differences (reDiff) obtained at each time point shown in Figure 9 and Figure S5. The blue histograms indicate the relative differences between $h_i$ and *h*, and the red histograms indicate the relative differences between $h_\theta$ and *h*.

Therefore, the observation-derived SBL heights determined from the virtual potential temperature profile $h_\theta$ in our study seem to be suitable for defining the SBL height. Furthermore, although $h_i$ performs slightly worse than $h_\theta$ during our study, we can still argue that $h_i$ is somewhat related to *h*.

In addition, we discovered that the vertical profiles we obtained in our study from all seven runs are inadequate. We further compared the wind profile data ($h_J$, $h_1$) of each run in Figure 12a, and the mean absDiff and mean reDiff between the *h* values calculated using the four derived methods, namely, $h_i$, $h_\theta$ $h_J$, and $h_1$, are shown in Figure 12b,c. The error bars represent the standard deviations of the mean observation-derived SBL heights calculated by each method. According to Figure 12a, all four methods, especially $h_\theta$, are in

good agreement with $h$ at most times. The results obtained using the wind profile data are very similar to $h$ with mean absDiff and reDiff values below 50 m and 25%, respectively. Specifically, a mean reDiff value lower than 23% is obtained for $h_J$. During Ep. 2, Ep. 3, and Ep. 4, $h_1$ is also consistent with $h$, although the one standard deviation (shown by the error bars in Figure 12c) with respect to $h_J$ is higher than 23%. In addition, for all four methods, using the virtual potential temperature method provides the best fit for $h$. Moreover, the standard deviation values of $h_\theta$ (shown by the error bars in Figure 12b,c) are the smallest. In addition, among all four methods, relatively good results are given by $h_J$ with mean absDiff and reDiff values of 48 m and 23%, respectively. In general, these findings suggest that the results of all four methods determined by meteorological variables, such as temperature and wind speed changes, are in good agreement with $h$, although the vertical profiles available are limited.

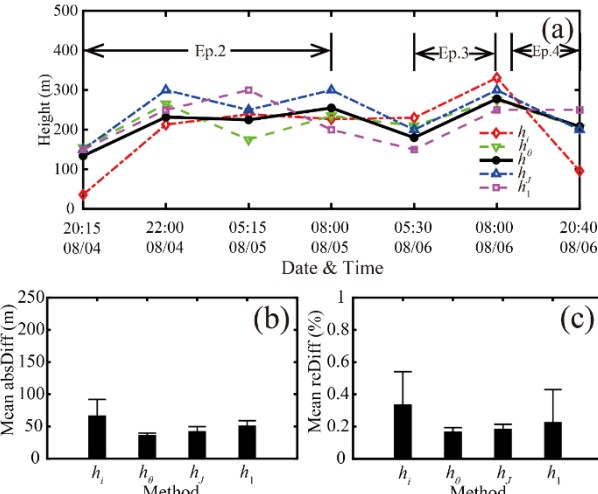

**Figure 12.** (**a**) Comparison among the four observation-derived SBL heights $h_i$, $h_\theta$, $h_J$, and $h_1$ (dashed lines) with the calculated SBL height $h$ (solid line) during three episodes: Ep. 2, Ep. 3, and Ep. 4. (**b**) Mean absolute difference (Mean absDiff) and (**c**) mean relative difference (Mean reDiff) values and corresponding one standard deviation (shown by error bars) obtained by using each method shown in Figures 8, 10 and S6 during the three episodes of Ep. 2, Ep. 3, and Ep. 4. In addition, it should be noted that Figure 12a is not simply a succession of time–height cross section; we simply superposed the episodes for convenience.

## 4. Conclusions

In this paper, we propose a composite iteration method to estimate the stable boundary layer height using wind profiles from Doppler lidar and turbulent fluxes from ultrasonic anemometer. By calculating a simple discriminant, continuous records can be classified as stationary or nonstationary according to the value of $\beta$. Before substituting values into the algorithm (Equations (2)–(6)), the averaging length ($L$ = 1 h) used to compute the fluxes must be chosen. Finally, these data are substituted into Equations (2)–(6), enabling $h$ to be determined. In addition, other methods for estimating the SBL height, namely, a prognostic equation ($\lambda(h)$) and three diagnostic equations ($\lambda_1$, $\lambda_2$ and $\lambda_3$), as well as observation-derived SBL heights ($h_J$: maximum wind speed height, $h_1$: zero wind shear height, $h_i$: temperature inversion height, and $h_\theta$: height at which 0.8 times the inversion strength appears for the first time) are presented, and the results are compared with the predicted SBL heights. The main results are as follows:

1.  $h$ is in good agreement with $h_i$ and $h_\theta$ obtained by radiosonde data, especially for $h_\theta$. A comparison of $h$ with the radiosonde-derived estimates demonstrates that $h_i$ presents a relatively poor result with mean absDiff and reDiff values of 72 m and 36%, respectively. $h_i$ and $h$ may be satisfactory but have minor differences. In addition, $h_\theta$

    shows the smallest mean absDiff and reDiff values (below 48 m and 22%, respectively). Moreover, with regard to the one standard deviation, $h_\theta$ shows the smallest values.

2. The heights derived from wind profiles ($h_J$ and $h_1$) also show good agreement with $h$. The SBL height derived from $h_J$ shows low absDiff and reDiff values below 50 m and 23%, respectively. However, for $h_1$, the mean relative error (46.0%) is twice as large as that for $h_J$.

3. The diagnostic formula of $\lambda_3$ fits the best with $h$ among the three diagnostic formulas, whereas the prediction equation is not applicable. Nevertheless, the diagnostic formulas of $\lambda_2$ and $\lambda_3$ are found to be appropriate, especially under extremely and moderately stable conditions. Furthermore, the performance of $\lambda_3$ presents the best results among all the dimensional scale height parameters. $\lambda_1$ shows less consistency with $h$, but under extremely stable conditions, all three diagnostic formulas provide good fits with $h$, especially those of $\lambda_2$ and $\lambda_3$. However, the prognostic equation of $\lambda(h)$ in our study is very unsatisfactory.

    Due to the limited amount of data (3 days), we used various methods for comparison. Comparing these different methods yields reasons to believe that the method proposed in this study can determine the thermodynamic SBL height once the wind profile (dynamic structure) and turbulent fluxes (thermal structure) are known. Moreover, our method includes the flux–profile relationship, which few studies have considered. The advantage of this method is that it can obtain continuous and accurate estimates of $h$ and is easier to popularize. Aerosols interact strongly with meteorological variables with the strongest interactions taking place in the ABL. Moreover, aerosols can increase atmospheric stability by inducing a temperature inversion as a result of both scattering and absorption of solar radiation, which suppresses dispersion of pollutants and leads to further increases in aerosol concentration in the lower ABL. Knowledge of the ABL is thus crucial for understanding the interactions between air pollution and meteorology. Our method of estimating SBL height is critical to improve the understanding of the air pollution and boundary layer interaction. However, there are still drawbacks; for example, in episodes of alternating daytime and nighttime hours, it is more complicated to determine $h$, so the height calculated is the residual height or the SBL height. In addition, we tried to use the flux–profile relationship to calculate the height of the convective boundary layer (CBL) over complex terrain but failed because the physical processes in the CBL are more complex than those in the SBL. Our future research will focus on other ways to determine the CBL height more accurately by using the flux–profile relationship. Moreover, because of the limited data and the vertical resolution of Doppler lidar (50 m), future experiments will involve long-term observations and HRDL, e.g., using HRDL with a vertical resolution of 1.5 m, to further validate the retrieval of the SBL heights from observation-derived measurements.

**Supplementary Materials:** The following are available online at https://www.mdpi.com/article/10.3390/rs13183596/s1. File S1: Introduction of UAT-2; File S2: The pre-processing of turbulent fluctuations; File S3: The computation of the dimensionless stability parameter; File S4: The method of Z test. Figure S1: Synoptic weather maps at sea level during the observation experiment from 5–6 August 2020; Figure S2: Three wind components and the velocity; Figure S3: A comparison of vertical profile of potential temperature and water vapor mixing ratio between Site A and Site B; Figure S4: Scatter diagram showing the values of wind profile $U\_real$ measured by FC-II and calculated $U\_cal$; Figure S5: Samples profiles shown by blue profiles and orange profiles of $U$ and $\partial U / \partial z$; Figure S6: Three times (05:30; 08:00; 20:40) were chosen that were marked in brown as background; Table S1: Operating parameters of FC-II and UAT-2; Table S2: Correlation coefficients between calculated observed rates of SBL and determined by the prognostic rate equations; Table S3: Classification of $h$ by using dimensionless stability parameter $\mu_0$.

**Author Contributions:** Conceptualization, H.C. (Hongyan Chen); methodology, H.C. (Hongyan Chen), H.S. (Haijiong Sun) and G.T.; software, H.S. (Haijiong Sun), C.S. and K.C.; validation, H.S. (Haijiong Sun), H.C. (Hongyan Chen), H.S. (Hongrong Shi), H.C. (Hongbin Chen); writing—review and editing, H.S. (Haijiong Sun) and H.S. (Hongrong Shi); supervision, H.C. (Hongbin Chen) and H.S. (Hongrong Shi). All authors have read and agreed to the published version of the manuscript.

**Funding:** This study is funded by the National Natural Science Foundation of China (Nos. 41805021 and 41775005) and the National Key R&D Program of China (Grant No. 2017YFA0603504).

**Institutional Review Board Statement:** Not applicable.

**Informed Consent Statement:** Not applicable.

**Data Availability Statement:** The data and code of this study are available from the authors upon request (sunhaijiong@mail.iap.ac.cn).

**Acknowledgments:** The authors wish to express their gratitude to all the engineers of Xianghe Station and Drone Team for their technical support in the field campaign. The authors would like to thank the Key Laboratory of Middle Atmosphere and Global Environment Observation, Institute of Atmospheric Physics, Chinese Academy of Sciences. The authors also would like to thank Fei Hu for providing FC-II wind data. The field experiment data were supported by the National Natural Science Foundation of China and the National Key R&D Program of China.

**Conflicts of Interest:** The authors declare no conflict of interest.

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
