# Peer review of "Evaluation of a Method for Calculating the Height of the Stable Boundary Layer Based on Wind Profile Lidar and Turbulent Fluxes"

_remotesensing, doi:10.3390/rs13183596_

Round 1

Reviewer 1 Report

- Authors proposed a new method to estimate the ABL height. The method takes into account thr turbulent vertical fluxes of momentum and remperature. I reccomend authors to give formulas for turbulent fluxes (line 186, as well as line 187 (parameters with_*)).

- Introduction. it would be useful if you could consider the following papers in your study:

[1] Zhang, H., Zhang, X., Li, Q. et al. Research Progress on Estimation of the Atmospheric Boundary Layer Height. J Meteorol Res 34, 482–498 (2020). https://doi.org/10.1007/s13351-020-9910-3

[2] Raghavendra Krishnamurthy et al. On the estimation of boundary layer heights: a machine learning approach / Atmos. Meas. Tech., 14, 4403–4424, 2021

https://doi.org/10.5194/amt-14-4403-2021

- Usually researchers consider either dynamic ABL or thermal ABL. In this study, the boundary layer may be called thermodynamic?

- The sentence “The daytime convective boundary layer height involves chaotic turbulent” is not quite clear. The authors talk about chaotic turbulence. Night turbulence is not chaotic? Perhaps the authors mean that the structure of turbulence is different and at night, due to stable stratification, turbulence is suppressed. Please rephrase this sentence.

-Line 118. Recomendation. You may show time series of the ultrasonic anemometer thermometer-II measurements.

-Also, the structure and height of the atmospheric boundary layer may be estimated from the measurements of the vertical profiles of the optical («thermal») turbulence estimated from recorded phase fluctuations [3,4].

[3] Osborn et al. Profiling the surface layer of optical turbulence with SLODAR / Monthly Notices of the Royal Astronomical Society, Volume 406, Issue 2, August 2010, Pages 1405–1408, https://doi.org/10.1111/j.1365-2966.2010.16795.x

[4] Shikhovtsev, A.Y., Kiselev, A.V., Kovadlo, P.G. et al. Method for Estimating the Altitudes of Atmospheric Layers with Strong Turbulence. Atmos Ocean Opt 33, 295–301 (2020). https://doi.org/10.1134/S1024856020030100

- Figure 4. What is the reason for the fact that the height of the boundary layer practically does not change during night.

- Figure 5. The authors describe the stability of the atmosphere (Extremly stable, very stable and so on). Please give the values of the parameter by which you evaluate the stability of the atmosphere, for example, in the form «Ri>0.25 — very stable».

-Figure 9 has bad quality. Please renew.

In general, I believe that the article is made at a good scientific level, contains new interesting results and can be published after the comments have been eliminated.

Reviewer 2 Report

General comments:

This paper presents the evaluation of an iterative method to estimate the stable boundary layer (SBL) heightusing wind profiles from Doppler lidar and turbulent fluxes from ultrasonic anemometer. Comparison with other methods (prognostic and diagnostic) for estimating the SBL height are presented. The results are compared with the predicted SBL heights are in good agreement with those derived from radiosonde data and wind profiles. The study is very interesting and is detailed so that I believe that it could be useful for the reader. The paper is suitable for the Journal. The manuscript is well structured and very good presented. The paper has some limitations due to the fact that only three days are discussed and authors should better explain why or how they believe that these results could be generalized. My advice to authors is to use more days in the future studies to be able make the conclusions more general. I would suggest to consider the paper for publication after minor revision.

Specific comments:

Line31: Equation of ?3 should be reformulate in order to improve the comprehension of the abstract.

Line 62: Some more recent references should be added.

Line 72 : Some more recent references should be added.

Figure 1: is not readable when printed.

Line 128 : The quintuple variance (5?2) method should be explained or references should be added.

Line 132: The methods used to calculate ?should be added in S2.

Line 148: The title of this subsection is not appropriate.

Line 186: Turbulent fluxes ? and ?should be defined correctly in the text.

Line 190: u* should be defined correctly in the text.

Line 220: The origin of the iterative algorithm used to relate the above to the wind profiles should be mentioned in the text.

Line 255: In Figure S3, correlation coefficient was R2=0.91 whereas in the text, it was 0.81. It is not clear.

Line 358: The authors should explain the method used to deduce hj(the height of maximum wind speed) in Type II characterized by a uniform or “flat” profile. How to determine the maximum speed from a uniform or “flat” profile

Line 437 : This sentence is not clear, please reformulate it

Reviewer 3 Report

Overview and General Impression
The manuscript (MS) presents composite iteration method for estimation of the height of the stable boundary layer using wind profiles from Doppler lidar and turbulent fluxes from ultrasonic anemometer. The method is implementation and extension of the semi-empirical proposal, described in a few articles with a leading author the recently passed away S. S. Zilitinkevich. As main merit of the MS, alongside the demonstrated deep knowledge of the considered matter, I would outline the comparison of the obtained result with the outcomes of other approaches.
The subject of the publication is relevant and relatively well (see below) motivated from the authors. It fits also good in the scope of ‘Remote Sensing’.
The MS suffers, however, from some weaknesses which have to be addressed before it becomes publishable.

Major remarks
May overall concern stems from the fact that the iteration procedure, which is the core of the MS, is  poorly described and, at least for me, remains fairly unclear. Any iteration procedure is, in fact, a sequence of finite number of successive steps with, known in advance, stop criterion. In the present MS there is no one equation, inclusive Eq. (6), row 221, which looks like iterative procedure (it could be expecting a formulae, expressing the linkage between adjacent steps). An excessive reformulation should be considered. Think about also for something like flow-chart from which the text, respectively the overall clearness could also benefit.
According my knowledge, the development of SBL is possible only under certain synoptic-scale background; not every nocturnal planetary/atmospheric BL could be regarded as SBL. Generally speaking, the weather should be enough calm, i. e. by anticyclonic or, at least, low-gradient (usually in the cold period of the year) conditions. Thus, a short description (motivation) of the synoptic condition during the field experiment should be considered.
The importance of the subject could be strengthened, emphasizing that high air pollution episodes near the sources (I. e. in the cities) are realized exactly in low turbulence conditions, most frequently in SBL.
The dimensionless stability parameter \mju_0 is too important, it is already persisting in Table 1, but not clarified. Table S3 is small enough and could be placed in the main text.
Finally, the study is some kind of survey of the existing methods of determination of the SBL-height, right? Thus, the MS could only benefit if include (in Section 3 or in the Conclusion), alongside the scientific evaluation (already done) concise judgment/recommendation for ‘most productive’ approach based on criteria as: applicability (hence, for example, the wind profile is strongly impacted near urbanized areas), best trade-off complexity/precision, etc.

Specific remarks
-r76, r78: the citation of Zilitinkevich et al. [30]
- Table S2 b) ‘Accuracy of wind direction’ persists twice, with two different values
- r203: Should be not the first term of RHS with subscript TN instead of R, similarly to the other two?

Round 2

Reviewer 1 Report

I recommend the manuscript for publication